# SIMULATE BEFORE ACT:
# MODEL-BASED PLANNING FOR WEB AGENTS

## ABSTRACT

Language agents have shown promising performance in automating web-based tasks, but the complexity and vast search spaces of real-world websites challenge reactive agents in identifying optimal solutions. While tree search agents offer enhanced exploration by interacting with actual websites, they often incur high costs, potential risks, and are challenging to implement for real-world websites. This paper explores a novel paradigm leveraging large language models' (LLMs) internal world models for planning in complex environments, presenting a middle ground between reactive agents and tree search agents. Results on two representative benchmarks, VisualWebArena and Mind2Web-live, demonstrate that our approach largely closes the gap between reactive agents and tree search agents, while maintaining efficiency and safety advantages. Notably, tree search can be considered as approaching an upper bound for our method, as it explores actual websites rather than simulations. This work opens new avenues for research into more effective and secure strategies for autonomous agents in complex, dynamic environments. It represents a step forward in improving upon reactive agents while approaching the performance of tree search methods, without incurring their implementation challenges and costs.[1]

## 1 INTRODUCTION

Planning, a cornerstone of artificial intelligence since its inception, continues to drive remarkable advancements in the field. From AlphaGo's (Silver et al., 2016) groundbreaking performance to recent investigations into scaling inference-time compute with large language models (LLMs) (Wang et al., 2024; Feng et al., 2023; Brown et al., 2024), these developments underscore planning's pivotal role in propelling AI capabilities to unprecedented heights. Notably, recent research demonstrates that augmenting LLMs with advanced inference-time algorithms, such as tree search (Yao et al., 2023; Hao et al., 2023), effectively improves performance on complex reasoning tasks compared to standard chain-of-thought (CoT) reasoning (Wei et al., 2022). These methods of scaling inference-time compute through planning algorithms enables LLMs to explore multiple potential solution paths, yielding more robust and accurate outputs.

However, translating these successes to complex real-world environments presents formidable challenges. This difficulty has contributed to a notable lag in research progress on planning in real-world scenarios. Specifically, the underlying dynamics or transitions of complex environments, such as the Web (Deng et al., 2023; Koh et al., 2024a; Pan et al., 2024b) or physical environments (Li et al., 2024; Shridhar et al., 2020), are often unknown or incomputable. This complexity prevents the direct use of search algorithms to find the best plans ahead of time. Consequently, most existing works adopt the *reactive agent* paradigm, where an action is directly executed based on the current observation at each step, without engaging in planning (Zheng et al., 2024; He et al., 2024; Cheng et al., 2024; Hong et al., 2024; Lai et al., 2024). However, this approach often leads to suboptimal outcomes due to insufficient exploration of the environment. To address the limitations of reactive agents, one might seek to conduct online exploration with the environment to implement *tee search* algorithms effectively. Yet, conducting online planning through real-time environmental exploration poses significant challenges in terms of efficiency and raises potential safety concerns (Koh et al.,

---

[1]Code and data will be released upon acceptance.

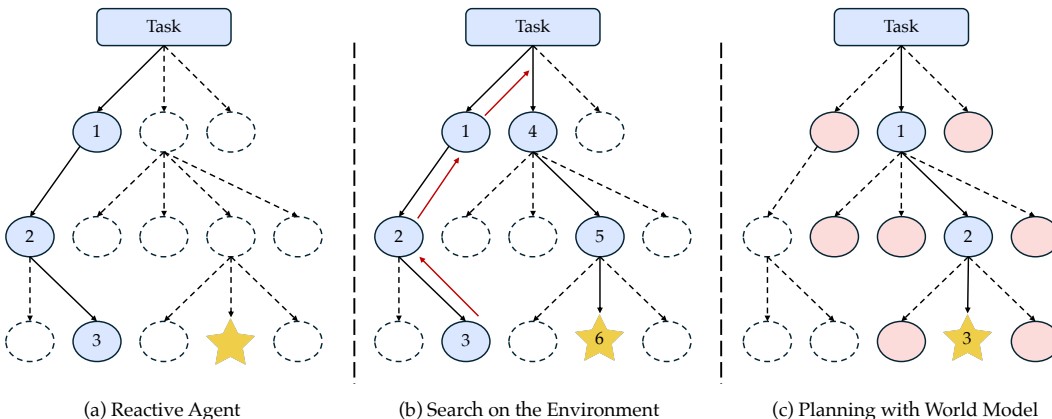

(a) Reactive Agent    (b) Search on the Environment    (c) Planning with World Model

Figure 1: Schematic illustrating various approaches for web agents formulated as a search problem over the webpage. Each node represents a webpage. Blue nodes are web pages actually visited. White nodes outlined with dashed lines are webpages that exist but are not visited. Pink nodes are not actually visited, but the agent can derive simulated observation with the world model. The Star node is the target the agent is required to reach. The number of nodes represents the order of being visited.

2024b; Putta et al., 2024). In addition, tree search may not always be viable in real-world environments due to the difficulty in backtracing; many actions in these contexts are irreversible, complicating the search process.

In this paper, we strive to find a middle ground between planning using fully online tree search and purely reactive agents based on CoT reasoning. Specifically, we build upon the hypothesis that *LLMs may inherently encode an internal world model of the environment*, as suggested by previous research (Hao et al., 2023; Kim et al., 2024). While these studies indicate LLMs' potential for world modeling, they have primarily focused on simple, constrained environments (*e.g.*, blocksworld). Our work pioneers the investigation of LLM-based world models in complex, real-world scenarios, particularly web navigation. We leverage LLMs as simulators to predict state transitions after executing actions, enabling model-based planning without actual environmental exploration (see Figure 1). To realize this, we devise a multi-stage framework for our model-based planning, comprising four stages: action proposal, self-refinement, simulation, and scoring. The latter two stages correspond to the transition model and reward model commonly used in world modeling. With the transition model, we simulate task execution within an imagined environment; with the reward model, we evaluate and score each simulated playout to guide the planning process. This synergy enables efficient exploration of potential action sequences without real-world interaction. Our planning algorithm's concept resembles model predictive control (MPC; Garcia et al. (1989); Kouvaritakis & Cannon (2016)), which effectively manages error accumulation in simulations: only the first action of the top-ranked simulated trajectory is executed, with new simulations generated from the resulting state. This process iterates until the model decides to terminate.

To validate the effectiveness of our model-based planning paradigm, we test it on open-ended web environments, where an agent is expected to automate diverse tasks over real-world websites that entail a tremendous search space (*e.g.*, booking a flight or looking for a specific product). We demonstrate the effectiveness of our proposed method on two representative benchmarks that support online interaction: VisualWebArena (Koh et al., 2024a) and Mind2Web-live (Pan et al., 2024b). Our model-based planning approach significantly outperforms the reactive agent on both datasets. Although its performance still falls short of tree search with actual interactions, it's important to note that the tree search agent can be viewed as an upper bound for our simulation-based method. Moreover, model-based planning offers superior flexibility compared to tree search, which is often inefficient and impractical to implement on real-world websites.

In summary, our paper presents a pioneering study on model-based planning utilizing the internal world models of LLMs in complex environments. As an initial exploration in this domain, we prioritize establishing the viability and potential of this paradigm over performance optimization.

Our novel approach effectively narrows the performance gap between reactive agents and tree search agents in real-world scenarios, demonstrating the promise of LLM-based world models for planning tasks. Through our experiments, we have not only validated the potential of this approach but also identified key limitations and challenges, providing valuable insights to guide future research in this emerging field.

## 2 RELATED WORK

### 2.1 WEB AGENTS

Driven by the goal of automating tedious and repetitive web-based tasks, web agents powered by (multimodal) language models have made substantial progress in various aspects. Benchmarks have evolved from MiniWoB++ (Shi et al., 2017; Liu et al., 2018) to WebShop (Yao et al., 2022) and WebArena (Zhou et al., 2023), offering increasingly realistic website simulations. VisualWebArena (Koh et al., 2024a) and Mind2Web (Deng et al., 2023) challenge models' ability to handle visual information and generalize across diverse tasks, websites, and domains.

**Reactive Agent.** Significant progress has been made to enhance the fundamental capabilities of web agents through both prompting closed-source models (Zheng et al., 2024; He et al., 2024; Deng et al., 2023) and training models using HTML and webpage screenshots (Lee et al., 2023; Gur et al., 2023; Furuta et al., 2023; Hong et al., 2024; Baechler et al., 2024). Additionally, models' abilities to ground web agent actions to elements have been improved through training on action-coordinate pair data (You et al., 2024; Cheng et al., 2024). Further advancements have been achieved by training on web agent trajectories, utilizing both human-annotated trajectories (Shaw et al., 2023; Hong et al., 2024; Deng et al., 2023; Lai et al., 2024) and synthesized exploration trajectories (Furuta et al., 2023; Song et al., 2024; Patel et al., 2024).

**Tree Search Agent.** Pan et al. (2024a) introduces a reward model based on GPT-4V, designed to provide both step-wise and trajectory-level rewards to guide inference-time search. Search Agent (Koh et al., 2024b) investigates inference-time search algorithms in interactive web environments, enabling explicit exploration and multi-step planning. In contrast to Search Agent, which employs a variant of best-first tree search, AgentQ (Putta et al., 2024) and WebPilot (Zhang et al., 2024) utilize Monte Carlo Tree Search (MCTS) as their primary search strategy.

While tree search on websites has demonstrated significant improvements, it still presents several limitations. First, the search process substantially increases inference time due to the need for extensive exploration, which is difficult to parallelize given its inherently sequential nature. Secondly, search necessitate backtracking to previous states. Although it is possible to implement in sandbox environments with heavy overhead by resetting the environment and storing the action sequence leading to a specific state (Koh et al., 2024b), is not feasible for real-world websites. Finally, search heightens the risk of destructive actions that may irreversibly alter the website's state, potentially causing harmful side effects.

### 2.2 WORLD MODELS

World models, a cornerstone of model-based reinforcement learning (Luo et al., 2024) since the introduction of Dyna by Sutton (1991), are typically trained on observed state transitions to predict future states and rewards. In addition to learned models, simulators with physical engines (Kolve et al., 2017; Puig et al., 2018) can also serve as world models. These world models enable efficient training through simulated experiences, reducing environmental interactions and improving sample efficiency (Ha & Schmidhuber, 2018). Beyond their role in training, researchers have explored the use of world models to facilitate planning (Pascanu et al., 2017; Schrittwieser et al., 2020). Fundamentally, world models in reinforcement learning often involve task-specific training, with a primary focus on enhancing data efficiency in the agent learning process.

In contrast to traditional world models in reinforcement learning, LLMs employed as world models primarily focus on facilitating decision-making in planning rather than training. This distinction leads LLM-based models to prioritize key task abstractions over the high-fidelity simulations typically required in reinforcement learning. Recent research has demonstrated the potential of LLMs

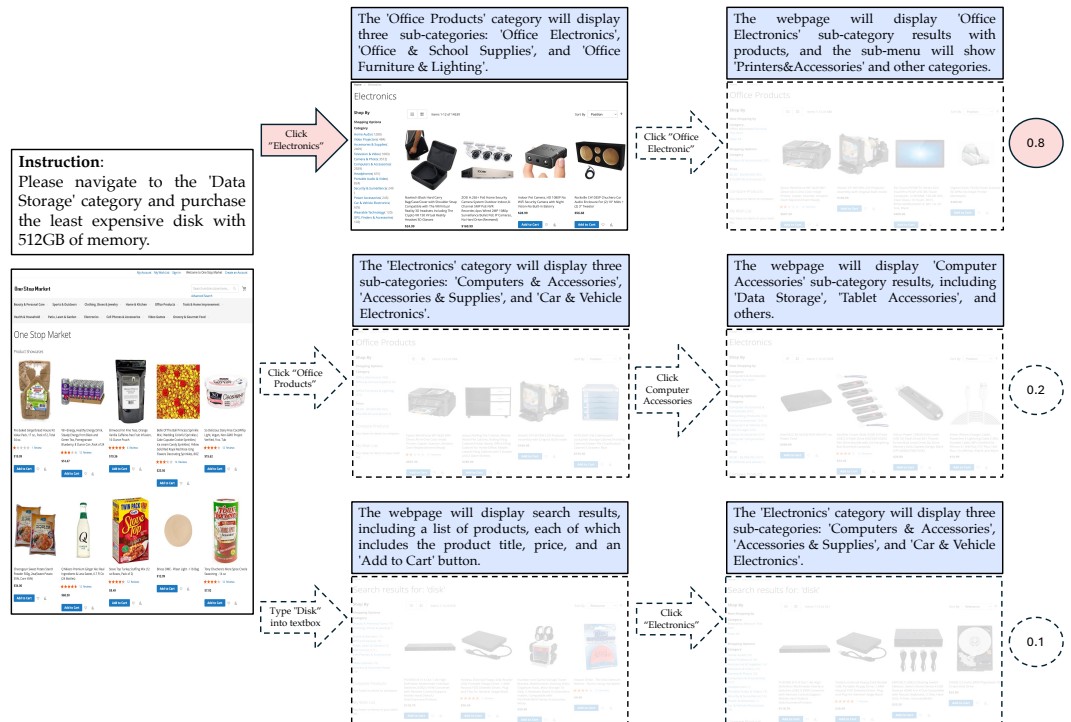

Figure 2: Illustration of our MPC-based planning using LLM simulation. The LLM simulates trajectories for three candidate actions: (1) *Click "Electronics"*, (2) *Click "Office Products"*, and (3) *Type "Disk" into textbox*. Through simulation and scoring of the simulated trajectories, the LLM identifies *Click "Electronics"* as the most promising action and executes it. The blue textboxes represent LLM-generated state change descriptions resulting from each simulated action. This example demonstrates a planning horizon of 2 steps.

as world models for simple environments, leveraging their encoded broad world knowledge (Hao et al., 2023; Kim et al., 2024). Our study aims to advance this field by investigating the capabilities of LLM-based world models in more complex real-world environments, specifically diverse websites.

## 3 METHOD

Web agents automating tasks in open-ended online environments face vast, complex search spaces where reactive or greedy strategies often fall short (Koh et al., 2024a). Conversely, tree search methods based on online interaction frequently incur high costs and raise safety concerns (Koh et al., 2024b; Putta et al., 2024). This paper pioneers a novel paradigm: harnessing LLMs' internal world models for virtual exploration through simulation. We posit that this new paradigm could establish a *middle ground* between reactive agents and tree search agents, potentially striking an optimal balance between accuracy and efficiency.

However, achieving good performance with model-based planning is not trivial, primarily due to the high complexity of real-world websites, which poses critical challenges for accurate simulation. Inaccurate simulations may diverge from the actual environment, potentially proposing actions within the multi-step simulation that are not actually available in the real environment. Poor simulation quality can also negatively impact the accurate assessment of action selection. To address these issues, we employ a Model Predictive Control (MPC)-based planning framework with LLMs' simulation, representing an initial effort to unlock the potential of model-based planning with LLMs for web automation.

---

**Algorithm 1:** LLM-based Model Predictive Control for Web Agents

---

**Input:** Task instruction $I$, initial observation $o_0$
**Output:** Sequence of actions $a_0, a_1, \ldots, a_T$
$t \leftarrow 0$;
**while** *task not completed* **do**
    $\mathcal{A}'_t \leftarrow \text{ActionProposal}(o_t)$;
    $\mathcal{A}_t \leftarrow \text{SelfRefinement}(\mathcal{A}'_t)$;
    **if** $|\mathcal{A}_t| = 1$ **then**
        $a_t \leftarrow$ the single action in $\mathcal{A}_t$;
    **end**
    **else**
        best_score $\leftarrow -\infty$;
        **for** $a_0^t \in \mathcal{A}_t$ **do**
            $o_0'^t \leftarrow o_t$;
            **for** $h \leftarrow 0$ **to** $H - 1$ **do**
                **if** $h > 0$ **then**
                    $a_h^t \leftarrow \hat{\pi}(I, o_h'^t)$;
                **end**
                $o_{h+1}'^t \leftarrow \hat{T}(o_h'^t, a_h^t)$;
            **end**
            score $\leftarrow \hat{R}(I, \{o_t, a_0^t, o_1'^t, a_1^t, \ldots, o_{H-1}'^t, a_{H-1}^t, o_H'^t\})$;
            **if** *score > best_score* **then**
                best_score $\leftarrow$ score;
                $a_t \leftarrow a_0^t$;
            **end**
        **end**
    **end**
    Execute $a_t$ and observe $o_{t+1}$;
    $t \leftarrow t + 1$;
**end**

---

## 3.1 MPC-BASED PLANNING

MPC (Garcia et al., 1989; Kouvaritakis & Cannon, 2016) is a classic control method that addresses model inaccuracies. It computes an optimal trajectory in simulation, but only implements the first action before re-planning with new observations. This step-wise planning approach is particularly well-suited to complex web environments, where obtaining a perfect world model is extremely challenging, if not impossible. Formally, given a natural language task instruction $I$, the planning algorithm seeks to find a trajectory of actions $a_0, a_1, \ldots, a_T$ that completes the task in the target environment. This environment is governed by a deterministic transition function $T : \mathcal{S} \times \mathcal{A} \to \mathcal{S}$, where $\mathcal{S}$ and $\mathcal{A}$ represent the state and action spaces, respectively. However, due to the complexity of real-world websites, $T$ is typically unknown. Moreover, the agent does not have direct access to the state $s_t \in \mathcal{S}$ of the environment. Instead, it must determine its actions based on an observation $o_t \in \mathcal{O}$ of the current state $s_t$. At each time step $t$, given a set of candidate actions $\mathcal{A}_t$ proposed from the environment, our MPC-based planning solves:

$$a_t = \arg\max_{a_0^t \in \mathcal{A}_t} \hat{R}(I, \{o_t, a_0^t, o_1'^t, a_1^t, \ldots, o_{H-1}'^t, a_{H-1}^t, o_H'^t\}) \tag{1}$$

subject to:

$$\begin{cases} o_0'^t = o_t \\ a_h^t = \hat{\pi}(I, o_h'^t), & h = 1, \ldots, H-1 \\ o_{h+1}'^t = \hat{T}(o_h'^t, a_h^t), & h = 0, \ldots, H-1 \end{cases} \tag{2}$$

Here, $H$ is the planning horizon (*i.e.*, simulation depth), $\mathcal{O}'$ is the simulated observation space, $\hat{T} : \mathcal{O}' \times \mathcal{A} \to \mathcal{O}'$ is the proxy transition function, and $\hat{\pi}$ is the policy function used in simulation. $\hat{R}$ evaluates the entire trajectory given the task instruction, with its output range in $[0, 1]$. This

simplification of evaluating the whole trajectory at once, rather than summing step-wise rewards, allows for easier implementation within the LLM framework. Functions $\hat{T}$, $\hat{\pi}$, and $\hat{R}$ can all be simulated using LLMs, leveraging the LLM's world knowledge for complex web environments. This process repeats at each time step, with a new optimization problem solved based on the latest observation.

Specifically, finding the solution to Equation 1 involves four stages using the LLM: *action proposal*, *self-refinement*, *simulation*, and *scoring* (see Figure 2). The action proposal stage maps from the current observation to a set of promising actions: $\mathcal{O} \rightarrow 2^{\mathcal{A}}$. This stage identifies promising actions to improve the coverage. After obtaining a set of candidate actions from the action proposer. The self-refinement stage refines the candidate actions obtained from the action proposer, determining which merit further exploration: $2^{\mathcal{A}'} \rightarrow 2^{\mathcal{A}}$. These two stages in combination generate $\mathcal{A}_t$. If $|\mathcal{A}_t|$ equals one, we execute the action directly without proceeding into the following stages. Otherwise, we further simulate a trajectory of length $H$ for each action in it using $\hat{T}$ and $\hat{\pi}$. Finally, each trajectory will be evaluated using $\hat{R}$ to obtain a numerical score, and the first action of the best-scored trajectory will be selected for execution.

Algorithm 1 outlines our planning framework. For the action proposal and scoring stages, we adapt the sampling approach introduced by Koh et al. (2024b), modifying their prompts to suit our context. Detailed prompts for each stage are provided in Appendix A.

## 3.2 STATE REPRESENTATION

A crucial aspect of our model-based planning lies in the design choice of $\mathcal{O}'$, *i.e.*, the state representation within simulation.[2] Ideally, one would aim to align the simulated observation space $\mathcal{O}'$ with the actual observation space $\mathcal{O}$. However, actual observations typically involve multimodal perception, including screenshots with Set-of-Marks annotations (Yang et al., 2023). This multimodal nature presents a challenge for current LLMs, which are limited to uni-modal generation, making it infeasible to use identical representations in simulations.

In designing $\mathcal{O}'$, we face unique constraints. Generating or processing visual elements is beyond the capability of text-based LLMs. Moreover, decoding complete HTML or accessibility trees within the simulation is computationally intensive and prone to errors, potentially introducing noise that could compromise the planning process. To address these challenges, we opt for a natural language description of the predicted state change as the new observation in simulation. This approach offers several advantages:

**Compatibility:** It aligns with the text-based nature of LLMs, enabling seamless integration within the simulation process.

**Flexibility:** Natural language can capture a wide range of state changes, from simple UI updates to complex interactions.

**Efficiency:** Textual descriptions are computationally less demanding than generating or processing complex structural representations.

**Relevance:** We can focus on describing the most pertinent changes, filtering out irrelevant details that might distract from the planning task.

Concrete examples of state change description simulated by LLMs can be found in Figure 2. We provide further insights in our design choice for $\mathcal{O}'$ in Section 5.1.

---

[2]In our simulation context, we use "state representation" and "observation representation" interchangeably. While these terms may have distinct meanings elsewhere, their boundaries often blur within LLM-based simulated environments.

# 4 EXPERIMENTS

## 4.1 SETUP

To properly test our planning framework's real-world performance, we use benchmarks with on-line evaluation, capturing the dynamic nature of web interactions. We focus on two representative benchmarks: VisualWebArena (VWA; Koh et al. (2024a)), which emphasizes a multimodal setting, and Mind2Web-live (Pan et al., 2024b), which operates with HTML by default. VWA comprises 910 tasks across three locally hosted websites: Shopping, Classifieds, and Reddit. In contrast, Mind2Web-live includes 104 tasks spanning 69 real-world websites. We adhere to the default settings of both benchmarks: for VWA, we use screenshots with Set-of-Marks prompting as the observation space, while for Mind2Web-live, we use HTML. For our LLM, we choose the most advanced multimodal LLM available, GPT-4o, as it best serves our aim to pioneer model-based planning with LLMs and explore the full potential of this envisioned paradigm. In our experiments, we empirically set the planning horizon $H$ to 1. A comprehensive analysis of this parameter is presented in Section 5.1.

To demonstrate the effectiveness of our proposal, we primarily compare our approach with two major baselines: the reactive agent and the tree search agent. For VWA, we evaluate against both baselines, using search agent (Koh et al., 2024b) as a representative implementation of tree search. However, for Mind2Web-live, we only compare with the reactive agent. Implementing a tree search agent for Mind2Web-live presents insurmountable challenges due to the complexity and dynamic nature of real-world websites. The lack of a controlled environment makes it virtually impossible to perform reliable backtracing, a crucial component of tree search.

Table 1: Results on VisualWebArena and Mind2Web-live. Our MPC-based planning approach effectively narrows the performance gap between the reactive baseline and tree search, even without additional exploration of the website. For Mind2Web-live, implementing tree search algorithms poses significant challenges due to the requirement for website backtracing, leading us to omit tree search performance metrics. This limitation further underscores the flexibility of our MPC-based planning method. We also include additional baselines (denoted by `gray` cells) to provide broader context. While these comparisons may not directly assess our core hypothesis, they offer valuable background for understanding our method's performance in the web navigation landscape. [†] We run the reactive baseline on VWA by ourselves because local hosting requirements may lead to hardware-dependent performance variations.

| Benchmark | Observation $\mathcal{O}$ | Method | Completion Rate | Success Rate |
|---|---|---|---|---|
| VisualWebArena | Screenshot+SoM | Gemini-1.5-Pro + Reactive (Koh et al., 2024a) | - | 12.0% |
| | | GPT-4 + Reactive (Koh et al., 2024a) | - | 16.4% |
| | | GPT-4o + Reactive (Koh et al., 2024a) | - | 17.7%[†] |
| | | GPT-4o + Tree Search (Koh et al., 2024b) | - | 26.4% |
| | | GPT-4o + MPC (Ours) | - | 23.6% (↑33.3%) |
| Mind2Web-live | HTML | GPT-4 + Reactive (Pan et al., 2024b) | 48.8% | 23.1% |
| | | Claude-3-Sonnet + Reactive (Pan et al., 2024b) | 47.9% | 22.1% |
| | | Gemini-1.5-Pro + Reactive (Pan et al., 2024b) | 44.6% | 22.3% |
| | | GPT-4-turbo + Reactive (Pan et al., 2024b) | 44.3% | 21.1% |
| | | GPT-3.5-turbo + Reactive (Pan et al., 2024b) | 40.2% | 16.5% |
| | | GPT-4o + Reactive (Pan et al., 2024b) | 47.6% | 22.1% |
| | | GPT-4o + MPC (Ours) | 49.9% | 25.0% (↑13.1%) |

## 4.2 MAIN RESULTS

**Effectiveness.** We present the overall performance results in Table 1. Our MPC-based planning approach demonstrates substantial improvements over the reactive agent on both VWA and Mind2Web-live datasets. Notably, on the VWA dataset, our proposed method achieves a 33.3% relative performance gain. Meanwhile, our proposal still underperforms the tree search baseline in terms of overall success rate. Despite these improvements, our approach still falls short of the tree search baseline in terms of overall success rate. It is important to note, however, that surpassing the accuracy of tree search is not the primary objective of our proposed method. In fact, the tree search agent can be considered an upper bound for our approach, as it engages in actual interactions rather

than simulations. On Mind2Web-live, MPC-based planning outperforms the reactive baseline by 2.9% (a relative gain of 13.1%), which is less significant than the improvement on VWA. However, it's worth noting that the Mind2Web-live dataset does not offer as much discriminative power, as evidenced by the minimal performance differences across multiple base LLMs shown in Table 1. The strong results on both VWA and Mind2Web-live indicate the effectiveness of our method across different observation settings.

We further conduct a more granular analysis comparing our proposed method to the reactive baseline on the VWA dataset across multiple dimensions. Table 3 demonstrates that our model-based planning approach consistently outperforms the reactive baseline across all websites and task difficulty levels, approaching the upper-bound performance achieved by tree search. On tasks of medium difficulty according to the official annotation by VWA, model-based planning even surpasses the performance of tree search (*i.e.*, 22.2% vs. 24.1%). Despite its promise, model-based planning still struggles with hard tasks in VWA that necessitate multistep simulations. The accuracy of simulations diminishes as the number of steps increases, presenting a significant challenge for handling hard tasks.

**Efficiency.** Another key advantage of model-based planning is its efficiency compared with tree search using actual explorations. As shown in Table 2, tree search requires approximately three times more steps than the baseline across all environments, whereas our method maintains comparable action steps. Notably, tree search introduces about ten times more wall clock latency due to the extra actions and backtracking, while the simulation overhead in our approach is minimal and can be further reduced with increased parallelization.

Table 2: Action steps and wall clock time on VWA.

(a) Number of Action Steps

| Steps | Reactive | Tree Search | MPC |
| --- | --- | --- | --- |
| Classifieds | 3.4 | 9.9 | 4.1 |
| Reddit | 5.1 | 13.6 | 5.2 |
| Shopping | 4.5 | 11.4 | 4.5 |

(b) Task Completion Wall Clock Time

| Seconds | Reactive | Tree Search | MPC |
| --- | --- | --- | --- |
| Classifieds | 68.3 | 749.2 | 183.6 |
| Reddit | 83.5 | 972.1 | 233.7 |
| Shopping | 87.7 | 785.7 | 179.4 |

Table 3: Success rate breakdown based on different dimensions. $\gamma = \frac{SR_{\text{mpc}} - SR_{\text{reactive}}}{SR_{\text{tree search}} - SR_{\text{reactive}}}$ measures the extent to which MPC narrows the gap between the reactive agent and the tree search agent.

(a) Websites

| Websites | Reactive | Tree Search | MPC | $\gamma$ |
| --- | --- | --- | --- | --- |
| Classifieds | 16.8% | 26.5% | 22.6% | 59.8% |
| Reddit | 15.3% | 20.5% | 18.6% | 63.5% |
| Shopping | 19.4% | 29.0% | 26.5% | 74.0% |

(b) Task Difficulty

| Difficulty | Reactive | Tree Search | MPC | $\gamma$ |
| --- | --- | --- | --- | --- |
| Easy | 28.8% | 42.3% | 37.4% | 63.7% |
| Medium | 16.4% | 22.2% | 24.1% | 132.8% |
| Hard | 10.7% | 14.9% | 12.7% | 47.6% |

## 5 DISCUSSION

### 5.1 STATE REPRESENTATION AND PLANNING HORIZON

Our MPC-based planning approach relies on two critical dimensions for simulation: the state representation and the planning horizon (*i.e.*, the simulation depth). To gain deeper insights into its effectiveness and limitations, we investigate how various configurations affect the final performance. Given the high computational cost of these experiments, we conduct this analysis using a subset of the VWA dataset, comprising 100 shopping tasks with officially annotated human trajectories.

In addition to the state change description used in our primary experiments, we explore alternative approaches where GPT-4o predicts either the HTML code or the accessibility tree of the resulting webpage within the simulation. For each of these state representations, we evaluate planning horizons of 1, 2, and 3 steps. As depicted in Figure 3, all three state representations significantly out-

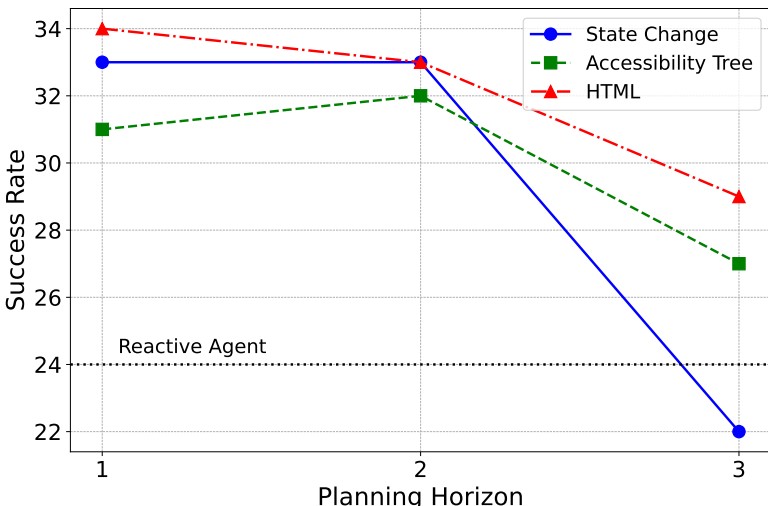

Figure 3: We demonstrate the performance on a subset of the VWA dataset, varying both the state representation within simulations and the planning horizon. Planning with long horizon with simulation remains challenging, regardless of the state representation employed.

perform the reactive baseline. However, their effectiveness diminishes as the planning horizon extends to 3 steps, indicating a common limitation in long-horizon simulation across these approaches. Specifically, the policy $\hat{\pi}$ within the simulation tends to hallucinate relevant actions for task completion, even when such actions may not exist in the current state predicted by $T'$. Notably, the state change representation exhibits the most pronounced performance degradation as planning horizons extend. This decline is particularly severe with a planning horizon of 3, where performance falls below that of the reactive baseline. This vulnerability stems from its implicit specification of available interactive elements on the current webpage, requiring the model to infer these elements by applying changes to the initial state. In contrast, HTML and accessibility tree representations provide explicit element information. Consequently, the state change approach is more susceptible to hallucination during extended simulations. Despite this limitation, the state change approach remains a viable choice given the current capabilities of LLMs. It matches the performance of HTML and accessibility tree representations for planning horizons less than 3 while consuming fewer output tokens.

## 5.2 ABLATION STUDY

To determine if the observed improvements come from specific parts of our model-based planning approach, we perform ablation studies on the simulation and self-refinement stages, using the same subset from Section 5.1. We pay special attention to the simulation stage, which is the core of model-based planning. One might argue that the improvement mainly comes from reranking candidate actions, regardless of whether this ranking is based on simulation. To test this idea, we conduct an experiment where we remove the simulation stage completely and instead ask the reward model to directly evaluate each candidate action. As shown in Figure 4, this modified approach does lead to some improvement over the baseline, but the gain is small and still falls well behind planning with simulation. These results confirm that the LLM-based world model simulation plays a crucial role in the planning process. Further-

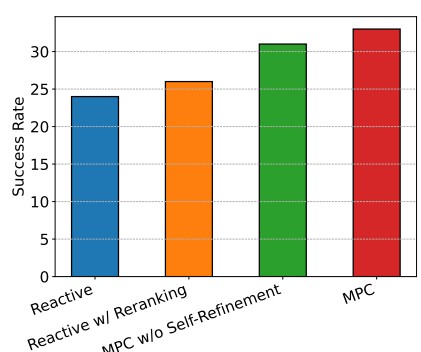

Figure 4: Ablation study on the simulation stage and self-refinement stage.

more, we observe a decrease in performance when removing the self-refinement stage. Upon closer examination, we find that this decline is primarily due to the self-refinement module's ability to effectively filter out less relevant candidate actions when the next optimal action is clear. In contrast, directly simulating all actions may introduce additional noise that can negatively impact performance.

## 6 CONCLUSION

In conclusion, our MPC-based planning approach demonstrates significant improvement over reactive baselines, effectively narrowing the gap with tree search methods across various web navigation tasks. This approach shows particular promise in scenarios where tree search implementation is challenging, highlighting its flexibility. However, to enable longer-horizon planning, future work must focus on enhancing the model's faithfulness to the real environment through targeted training. Additionally, exploring more sophisticated state representations within simulations presents a promising avenue for further performance gains.

## LIMITATIONS

Our study, as a pioneering exploration of MPC-based planning with LLMs for web navigation, naturally comes with certain limitations that pave the way for exciting future research directions:

**Simplicity of Planning Algorithm.** In this preliminary work, we deliberately employed a straightforward planning algorithm to demonstrate the core potential of our approach. While effective, this simplicity leaves ample room for future enhancements. More sophisticated planning techniques, such as Monte Carlo Tree Search (MCTS), could be integrated to further improve performance. As a foundational study, our focus was on establishing the viability of the concept rather than optimizing every aspect of the system. This strategic choice allows future research to build upon our findings and explore more advanced planning strategies within the framework we've established.

**Computational Cost.** Our current implementation, utilizing state-of-the-art models like GPT-4o, incurs significant API costs (approximately $1 per task on VWA). This cost reflects our prioritization of exploring the full potential of LLM-based planning without immediate constraints. For practical applications, future work could investigate cost-effective alternatives such as fine-tuning specialized models for simulation tasks. Our approach of using the most advanced available model serves as an upper bound, demonstrating the maximum potential of this paradigm. This sets a benchmark for future optimizations that balance performance and efficiency.

These limitations underscore the nature of our work as a proof-of-concept, opening up numerous avenues for future research and optimization. By establishing the foundational potential of MPC-based planning with LLMs, we have laid the groundwork for a new paradigm in web navigation, inviting further innovations that can refine and extend our approach.

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

# A  PROMPTS FOR FOUR STAGES IN MPC-BASED PLANNING

## A.1  ACTION PROPOSAL

---
**Action Proposal**

You are an autonomous intelligent agent tasked with navigating a web browser. You will be given web-based tasks. These tasks will be accomplished through the use of specific actions you can issue.

Here's the information you'll have: {Web Information}

The user's objective: {Task Objective} This is the task you're trying to complete.

The current web page screenshot: {Web Page Screenshot Image} This is a screenshot of the webpage, with each interactable element assigned a unique numerical id. Each bounding box and its respective id shares the same color.

The observation, which lists the IDs of all interactable elements on the current web page with their text content if any, in the format [id][tagType][text content]. tagType is the type of the element, such as button, link, or textbox. text content is the text content of the element. For example, [1234][button]['Add to Cart'] means that there is a button with id 1234 and text content 'Add to Cart' on the current web page. [][StaticText][text] means that the element is of some text that is not interactable.

The current web page's URL: {Web URL} This is the page you're currently navigating.

The open tabs: {Previous Tabs} These are the tabs you have open.

The previous action: {Previous Action} This is the action you just performed. It may be helpful to track your progress.

The actions you can perform fall into several categories:

Page Operation Actions:
 - click [id]: This action clicks on an element with a specific id on the webpage.
 - type [id] [content]: Use this to type the content into the field with id. By default, the Enter key is pressed after typing unless press_enter_after is set to 0, i.e., type [id] [content] [0].
 - hover [id]: Hover over an element with id.
 - press [key_comb]: Simulates the pressing of a key combination on the keyboard (e.g., Ctrl+V)
 - scroll [down] or scroll [up]: Scroll the page up or down.

Tab Management Actions:
 - new_tab: Open a new, empty browser tab.
 - tab_focus [tab_index]: Switch the browser's focus to a specific tab using its index.
 - close_tab: Close the currently active tab.

URL Navigation Actions:
 - goto [url]: Navigate to a specific URL.
 - go_back: Navigate to the previously viewed page.
 - go_forward: Navigate to the next page (if a previous go_back action was performed).

Completion Action:
 - stop [answer]: Issue this action when you believe the task is complete. If the objective is to find a text-based answer, provide the answer in the bracket.

Homepage:
If you want to visit other websites, check out the homepage at http://homepage.com. It has a list of websites you can visit. http://homepage.com/password.html lists all the account name and password for the websites. You can use them to log in to the websites.

To be successful, it is very important to follow the following rules:
 1. You should only issue an action that is valid given the current observation
 2. You should only issue one action at a time.
 3. You should follow the examples to reason step by step and then issue the next action.
 4. Generate the action in the correct format. Start with a *"In summary, the next action I will perform is"* phrase, followed by action. For example, *In summary, the next action I will perform is* click [1234].
 5. Issue stop action when you think you have achieved the objective. Don't generate anything after stop.

---

## A.2 SELF-REFINEMENT

---

**Self-Refinement**

You are assiting a web navigation agent to help a human user navigate a website to complete a task. Given the user's intent, the action history, and the current state of the webpage, the agent has proposed a set of candidate actions to take at the current step.

Your role is not to determine a best action for the agent at this step, but to filter out the actions that are very likely not relevant or helpful for the agent to accomplish the task.

Please select all actions that you think that could possibly lead the agent to accomplish the task. It's important to note that to accomplish a task, the agent will execute a sequence of actions. So the action to take at this step does not have to immediately lead to the completion of the task. You should select any action that could be relevant for the agent to take in the current state of the webpage. Try to be as thoughtful and comprehensive as you can! Don't miss any possible action. If there is one action that is clearly the best, and all other actions are clearly not very relevant, you can only select one action. Please do this sparely, since some actions may be helpful in a longer horizon.

A action should be included as long as it could be relevant to the task, even if it may not be the most direct action to take at this step!! Some relevant actions might seem indirect at the first glance, but could be helpful in a longer horizon. Please also include those actions.

Please at least select one action.

**\*IMPORTANT\***
Format your response into two lines as shown below:

Thoughts: `<your thoughts and reasoning process>`. You must explicitly evaluate each action one by one and imagine whether it could be relevant to the task following the format: `action:...  rationale:...`

Selected actions: `id0;id1;aid2;...` (please return the index of the action in the candidate actions list, starting from 0. Don't output the action description itself. Separate the indices with semicolons. Do not add spaces or any other characters between after the semicolons.)

Action History: {`last_actions_str`}

Current URL: {`current_url`}

The images corresponding to the user intent are shown in the FIRST {`len(intent_images)`} images (before the User Intent).

The last {`len(screenshots)`} snapshots of the agent's trajectory are shown in the LAST {`len(screenshots)`} images. The LAST IMAGE represents the current state of the webpage.

Proposed Action: {`action_descriptions`}

---

## A.3 WORLD MODEL

---

**World Model**

You are an agent that predicts the effect of an action on a webpage. You will be given a screenshot of a webpage, a sequence of actions and state changes applied to the initial screenshot, and an operation to perform on the webpage. You are required to predict the new changes that will occur on the webpage after the operation is performed, such as the appearance of new elements, the disappearance of existing elements, or changes in the content of existing elements. The operation type and the element to operate will be provided in the prompt. Directly output `State changes:...` and don't output anything else. Try to be as comprehensive and detailed as possible.

Based on the initial screenshot and the changes to the webpage, please predict the changes after action:

---

## A.4 REWARD MODEL

> **Reward Model**
>
> You are an expert in evaluating the performance of a web navigation agent. The agent is designed to help a human user navigate a website to complete a task. Given the user's intent, the agent's action history, the current state of the webpage, your goal is to decide **whether the simulated steps by the agent indicate a successful execution of the user intent**. In particular, if the predicted state (i.e., the current state represented by the last image plus all the predicted changes so far) corresponds to a successful final state. If it is a failure but it looks like the simulated steps are on the right track towards success, you should also output as such. Note that, in the simulated steps, all the state changes are predicted by the agent's world model, and they may not actually be faithful to the real website interactions (e.g., some proposed actions may not be avaiable in a realistic website). You should also account for this in your evaluation (e.g., if the predicted state changes are not reasonable then it's probably a failure).
>
> **\*IMPORTANT\***
>
> Format your response into two lines as shown below:
>
> Thoughts: `<your thoughts and reasoning process>`
>
> Status: `"success"` or `"failure"`
>
> On the right track to success: `"yes"` or `"no"`

