# OpenReview forum: "Simulate Before Act: Model-Based Planning for Web Agents"
_ICLR.cc/2025/Conference — ICLR 2025 Conference Withdrawn Submission_

### Official Review · Reviewer_5CFa · 2024-10-29

**Soundness:** 3
**Presentation:** 3
**Contribution:** 2
**Rating:** 5
**Confidence:** 3

**Summary:**

The paper presents an algorithm for improving web agents by incorporating planning into their execution. Instead of relying on full tree search or backtracking, the authors propose using the LLM's ability to predict the outcome of an action through a textual description, referred to as the "internal world model". Experiments show that, with a planning horizon of H=1, the agent evaluated in this way outperforms the reactive agent baseline.

**Strengths:**

The idea is simple and effective, and the paper is clear and easy to understand. The improvement demonstrated in the experiments is significant. In terms of novelty, the main idea is a straightforward yet novel combination of existing tools, which is fine. While it is possible that similar approaches have been proposed previously, I am not familiar enough with this area to give any specific example.

**Weaknesses:**

I like the simplicity of the main idea. However, I was a bit disappointed to learn that you're fixing the horizon to H=1. If I understand correctly, this means you essentially use the model to simulate the outcome of each relevant action and commit to the best one. While this is technically planning, it is borderline planning (just a bit more than scoring state-action pairs). You mention that "our planning resembles MPC, which effectively manages error accumulation...," which is generally true, but such claims are trivialized when the horizon is a single step. In this case, phrases like "pioneering study" also seem a bit farfetched.

What is the main issue that makes longer horizons unhelpful? Is it related to the hallucination issue you describe? I think the paper would benefit strongly from further development of the representations to make them reliable enough to increase the effective planning horizon. Otherwise, the approach may struggle to scale.

Section 3.2 describes the representations and lists their advantages, but it would also be a good place to discuss their limitations. To clarify, I'm not asking for a critique, as any choice of representations is a tradeoff. However, discussing this tradeoff here would be both interesting and valuable for future development.

The text, while mostly clear, needs improvements. Some sections are overly wordy (e.g. introduction), which makes it harder to focus on the important parts. Some paragraphs are simply unnecessary (most clearly lines 522-525). Additionally, some paragraphs sound generated by an LLM, which isn't an issue if they are concise (though that’s not always the case here). I recommend a careful proofreading.

It also seems that parts of Figure 2 are misplaced (e.g., "Electronics" is swapped with "Office Products").

**Questions:**

- Is the tree search agent (upper bound) equivalent to your MPC, only using the perfect model instead of the LLM-based simulation? It's not clear, but I think it's not. Such an agent sounds like a true upper bound. Please discuss that.

- Please specify which parts (i.e. components, algorithms, prompts, etc.) are shared by the main methods you compare (reactive, MPC, tree search).

- In Section 4.2 you show that MPC exceeds the performance of tree search. How can it be the case?

**Details Of Ethics Concerns:**

No concerns.

---

### Official Review · Reviewer_rDWR · 2024-11-04

**Soundness:** 3
**Presentation:** 3
**Contribution:** 3
**Rating:** 8
**Confidence:** 4

**Summary:**

The paper introduces a model-based planning approach for web agents, which combines elements of reactive agents and tree search methods to enable autonomous agents to navigate complex web environments efficiently. The authors propose using large language models (LLMs) to simulate actions in a virtual environment, minimizing real-world interaction and associated risks. They employ MPC to implement this framework, with the agent simulating action sequences before executing only the first action of the most promising sequence. Testing on benchmarks like VisualWebArena and Mind2Web-live, the method shows performance gains over-reactive agents, although it falls slightly short of the upper bounds achieved by tree search.

**Strengths:**

1. The approach is innovative in using LLMs as internal world models for web navigation, which addresses the limitations of both purely reactive agents and full tree search methods. This middle-ground approach leverages the predictive capabilities of LLMs for complex tasks, presenting a new application of LLMs in a web automation context.
2. The paper presents thorough empirical validation across two benchmarks, demonstrating the effectiveness of the proposed model against standard baselines. The inclusion of ablation studies and sensitivity analyses further enhances the robustness of the findings.
3. The paper is well-structured, with clear explanations of each step in the MPC framework. Figures and tables effectively illustrate experimental setups, results, and comparisons with baselines.
4. The proposed approach is highly significant for the development of web agents, as it reduces the computational and risk constraints associated with online planning. The method could lead to safer, more efficient web automation tools and potentially influence future models for other complex environments.

**Weaknesses:**

1. While the model shows promising results, its reliance on LLM-generated simulations raises questions about the fidelity of these simulations to actual web environments. As noted in the paper, inaccuracies in the LLM-based simulation may lead to suboptimal action choices, especially for tasks requiring long-horizon planning.
2. The authors acknowledge that the method incurs high computational costs, particularly when using state-of-the-art models like GPT-4o. This limits practical applicability, and future optimizations for cost reduction could improve usability.
3. The approach struggles with tasks that require multi-step planning, as performance deteriorates with longer horizons. This limitation is apparent in high-difficulty tasks on the VisualWebArena benchmark, highlighting the need for more advanced multi-step planning within the simulation framework. Do you have any modifications on how to improve this?
4. The results on Mind2Web-live are less compelling than on VisualWebArena, potentially due to the complexity and variability of real-world websites. This indicates that the approach may be more suitable for controlled environments and could face challenges in broader real-world applications. (I am slightly unfamiliar with the literature/benchmarks used for web-based agents)

**Questions:**

1. Could alternative state representations or intermediate reward structures improve the accuracy of the simulated state transitions in longer-horizon planning tasks?
2. Given the high computational costs, have the authors considered fine-tuning or using smaller, task-specific LLMs as a substitute for GPT-4o in the model-based planning framework?
3. The paper mentions that tree search serves as an upper bound for the proposed method. Could further integration of tree search techniques within the MPC framework enhance performance without compromising efficiency?
4. Since multi-step simulations present challenges, would it be feasible to integrate a mechanism for error correction in simulated trajectories? This might help mitigate the performance degradation observed in longer planning horizons.

Minor typos:

Line 50: “Tee Search” instead of “Tree Search”

Line 143:  "search necessitate backtracking to previous states." "search necessitate" should be "search necessitates."

---

### Official Review · Reviewer_qrxY · 2024-11-04

**Soundness:** 2
**Presentation:** 2
**Contribution:** 1
**Rating:** 1
**Confidence:** 4

**Summary:**

The presented work explores the usage of pre-trained LLMs to plan in complex web environments, aiming to balance between reactive agents and tree search agents. However, several issues need attention.

**Strengths:**

The work showcases a novel use case of pre-trained large language models for planning in complex web environments.

**Weaknesses:**

1. The planning process lacks adequate explanation, particularly regarding self-refinement and how the methodology differs from reactive methods if it doesn't involve backtracking.
2. The effectiveness of the results is questionable due to a planning horizon limited to one step.

**Questions:**

1. What do you mean by internal world models of LLMs? This is vague and has no strong meaning, do you mean a pre-trained language model?
2. Figure 2 has a mistake. Office products and Electronics categories in the search tree do not match with the arrows. Please update and change the score accordingly. The score does not match the example narrative.
3. The authors need to better explain and articulate the planning process involved (Section 3.1).
4. What is self-refinement in Figure 2?
5. Does the suggested MPC methodology perform back-tracking like tree methods? If not how is this different from reactive methods?
6. In Section 3.2, while discussing the ideal representation of observation space representation, the authors mentioned that HTML is computationally intensive, however in Section 4.1, for Mind2Web-live benchmark, authors have used HTML for O. This is contradictory.
7. The discussion in Section 3.2 initially suggests that the authors are introducing innovative steps to process states into natural language. However, it becomes apparent that the approach mainly involves utilizing trajectories from benchmarks as the basis for state representation, rather than employing a new processing approach.
8. It is unclear how the results demonstrate effectiveness, given that the authors specified a planning horizon of only 1 (Section 4.1 setup). This needs discussion in the paper. A "planning horizon of 1" in the planning context means that the model or agent only looks one step ahead when making decisions.

**Minor Comments:**
1. SoM is directly used in Table 1 without defining earlier. This should be defined in section 3.2.
2. Authors need to correct typos and grammatical mistakes. eg., Page 1, ln 50, ‘tree’ is spelled wrong.

---

### Official Review · Reviewer_FEv1 · 2024-11-04

**Soundness:** 3
**Presentation:** 3
**Contribution:** 3
**Rating:** 5
**Confidence:** 3

**Summary:**

This paper proposes a method for controlling a web agent. The main novelty is to use an LLM to simulate the execution agent’s actions outcome, and use these simulations to choose more effectively the agent’s next actions. This is done by simulating the execution of the actions the agent can choose, observing the “obtained” reward, and choose the action that maximizes this reward. The authors explored different planning horizons, i.e., different lengths of simulated trajectories, but the results show that one-step lookahead is best.  They call this approach MPC (due to its similarity to MPC in control theory). MPC also include an action selection and refinement step, where the latter asks the LLM to filter out some actions. Experimental results on two domains show that a one-step lookahead, which evaluate an action based on the reward obtained in the simulated next state, is better than a reactive agent.

**Strengths:**

1.	The web agents domain is interesting, relevant, and challenging.
2.	Exploring the use of LLMs to simulate executions for planning is an exciting direction.
3.	The experimental results show the proposed MPC algorithm is better than the Reactive agent and faster than the Tree Search agent.
4.	I like the action refinement stage. Also, the ablation study shows much of the benefit is attributed to it.
5.	A nice discussion on different ways to represent the states in this domain.
6.	The ablation study is very helpful.

**Weaknesses:**

1.	I am not sure the proposed project is mature enough. Since the objective is to use LLMs to plan for web agents, I would expect the planning component to be richer. Currently, it’s not really planning but more a one-step lookahead.
2.	I find the depiction of the results somewhat misleading with respect to the comparison of the tree search and MPC results. As I see it, the Tree Search is most of the time better, and the advantage of MPC over it is only runtime and number of simulations. This is not said clearly enough. In fact, in Table 1 the authors write that there is a 33% increase over the baselines but this is not true when considering Tree Search.
3.	In the proposed algorithm, only one simulation is done per action. This makes sense when the planning horizon is 1, but when the planning horizon this is not very reasonable.
4.	I like a lot section 3.1, which clarified to me finally the exact setting you were dealing with. However, I am not sure that viewing the environment as deterministic (see first paragraph of 3.1) is a valid assumption in the web agents domain.

Minor issues:
-	“As an initial exploration in this domain, we prioritize establishing the viability and potential of this paradigm over performance optimization” – I could not fully understand this sentence.
-	“... storing the action sequence .... not feasible for real-world websites.” – why? Storing sequences of actions isn’t easy to do?
-	“… LLMs employed as world models primarily focus on facilitating decision-making in planning rather than training.” – I am not sure what you mean in this sentence by “planning” and “training” (I’ve worked on both planning and training, but I still can’t make sense of this sentence).
Suggestions:
-	More information on what Web Agents are supposed to do with some examples would help section 2.1.

**Questions:**

1.	The authors write that the Tree Search results can serve as an upperbound. I do not see why this is true. I hope the authors can elaborate on this.
2.	I wonder if there aren’t any stronger baselines than the reactive agent for this type of web agents?
3.	Can you provide more details on the baseline Reactive agent? Did it also use the same set of actions? how did it evaluate these actions?
4.	How is this work novel compared to others that have used LLMs to simulate executions for planning? Or is this the first work to do so?
5.	Page 6. “This stage identifies promisting actions to improve the coverage.”  - what is “coverage” in this context?
6.	Why there are no completion rate results in VWA?

---

### Note · Authors · 2024-11-15

**Comment:**

We sincerely appreciate the thoughtful feedback from all reviewers. We acknowledge that the submitted version may need further refinement, so we have decided to improve it further and submit a polished version at a later date.

Here we briefly address some concerns from the reviewers:
> 1. Planning horizon is limited to one step

First, we would like to point out that all different state representations consistently outperform the reactive baseline across all horizon settings (see Figure 3).
Specifically, there is a slight improvement from planning horizon 1 to 2 for the state change description and accessibility tree.
We chose horizon 1 for our main experiments because it performs comparably to horizon 2 while being significantly less costly.
Also, we want to highlight that achieving better performance with longer horizons is not a major goal of this work.
Instead, our aim is to demonstrate the viability and potential of using LLMs for model-based planning in complex environments, a goal we have successfully achieved in this paper.

> 2. Inaccurate claim that tree search is an "upper-bound" of our method

We thank the reviewers for identifying this issue. We agree that the claim lacks rigor. The core problem is that tree search is not always feasible on real-world websites due to irreversible actions, despite its slightly better performance on VWA. Our model-based planning offers greater flexibility and, importantly, can complement tree search methods, for example, by using simulation to enhance the value function in tree search.

>3. The superiority of state change description over HTML/accessibility tree

We agree that we should make no claims about the strict superiority of state change description over HTML or accessibility tree. Also, this is orthogonal to our main message. As a pioneering study on LLM-simulated world models, our goal is to show that LLM's general knowledge about the websites can indeed help the performance of web agents, regardless of the representation (see Fig 3).

Unfortunately, we did not make these points clear in our submitted version. Therefore, we have decided to withdraw the submission and further refine the writing.

**Withdrawal Confirmation:**

I have read and agree with the venue's withdrawal policy on behalf of myself and my co-authors.